# OpenReview forum: "Sparks of Cooperative Reasoning: LLMs as Strategic Hanabi Agents"
_ICML.cc/2026/Conference — ICML 2026 regular_

### Official Review · Reviewer_whGx · 2026-03-07

**Soundness:** 3
**Presentation:** 4
**Significance:** 3
**Originality:** 2
**Overall Recommendation:** 5
**Confidence:** 3

**Summary:**

This paper introduces the first large-scale study of 17 different LLMs as Hanabi playing agents and releases two datasets with reward annotation. Notably, this is studied across 2 to 5 player settings, and varying levels of prompting templates are provided to facilitate reasoning from basic information to multi-turn working memory. Experiments show that compared to specially trained RL agents, LLM agents may pose more robust reasoning capabilities.

**Compliance With Llm Reviewing Policy:**

Affirmed.

**Key Questions For Authors:**

- Are there preliminary experiments done on having LLM agents and RL agents on the same team?

**Limitations:**

yes

**Strengths And Weaknesses:**

**Strength**
- Impressive and thorough effort in getting 17 different LLMs to play across 2-5 player settings of Hanabi. The accompanying datasets are sure to be useful as well, especially as they contain reward information, unlike previous datasets.
- Novel experiment protocol of prompting LLMs in achieving different levels of reasoning capabilities via Watson, Sherlock, and Mycroft templates.
- Great ablation studies, particularly on crossplay and generalization. The former could have big inspirations for the zero-shot coordination community, while the latter is unexpected and interesting.

**Weaknesses**
- 10 games per game setting is rather than limited. Althougth this is not trivial by any means.
-

---

> ### Author Rebuttal · Authors · 2026-03-30
>
> We thank the reviewer for their positive assessment of our work and are glad they found our work **"impressive and thorough"** with a **"novel experiment protocol"**, useful dataset contribution, and **"great ablation studies"** that potentially have **"big inspirations"** for zero-shot coordination between LLM agents. Below we address the weaknesses (W) and key questions (KQ) raised in the review.
>
> 1. **W1:** We thank the reviewer for bringing up this point and further elaborate on its non-trivial nature. Our evaluation of 10 games per player configuration is consistent with, or exceeds, the standard practice used in prior Hanabi benchmarks (*Yao et al., 2025; Chen et al., 2024*). We acknowledge this per-setting sample size is limited, and therefore follow the recommendations of *Agarwal et al. (2021)* who demonstrate that interquartile mean (IQM) with 95% confidence intervals is more robust to outliers than mean-based aggregation in low-sample evaluation regimes. We **replicate all results from the main paper with IQM** with 95% CI  in Appendix E.1 (Figures 9–15) and show that **the trends reported in the main paper with standard deviation hold under robust statistical aggregation** despite the per-setting sample size, consistent with the conclusions in the main paper.
>
> 2. **KQ1:** We do not conduct hybrid LLM–RL agent experiments in this work. The main challenge is a **fundamental representation mismatch**: classical Hanabi RL agents such as BAD (*Foerster et al., 2019*), SAD (*Hu & Foerster, 2020*), and OBL (*Hu et al., 2021*) operate on compact vectorized observations and learn implicit signaling conventions that are not directly interpretable to LLM agents, and vice versa. Placing such agents on the same team would therefore require a **non-trivial translation layer between the two representations**. Moreover, prior work has shown that classical RL Hanabi agents, despite strong self-play performance, often **do not generalize well to cross-play settings** (*Hu et al., 2020*), unlike the LLM agents studied in our work. Our cross-play results in Section 5.1 provide encouraging initial evidence for heterogeneous cooperation: mixing LLM agents of different capability levels yields relatively smooth performance interpolation, rather than the sharp degradation often observed in classical RL cross-play. This suggests that a hybrid LLM–RL team may exhibit similarly intermediate behavior, where a stronger LLM partner could potentially improve over the standalone cross-play performance of a classical RL agent. We leave a systematic empirical study of hybrid LLM–RL cross-play, as well as broader multi-agent collaboration settings discussed in Appendix A, to future work.
>
> We sincerely thank the reviewer for their time and positive feedback on the contributions of our work. *We are happy to have further discussion in the next couple of days to clarify any other concerns and further improve the manuscript.*
>
> ### References
> ---
> Foerster et. al 2019 - Bayesian action decoder for deep multi-agent reinforcement learning - https://arxiv.org/abs/1811.01458
>
> Hu et. al 2021 - Off Belief Learning - https://arxiv.org/abs/2103.04000
>
> Hu and Foerster 2019 - Simplified Action Decoder for Deep Multi-Agent Reinforcement Learning - https://arxiv.org/abs/1912.02288
>
> Chen et. al 2021 - Evaluating large language models trained on code - https://arxiv.org/pdf/2107.03374
>
> Yao et. al - SPIN-bench: How well do LLMs plan strategically and reason socially? - https://arxiv.org/pdf/2503.12349
>
> Agarwal et. al 2021 - Deep Reinforcement Learning at the Edge of the Statistical Precipice - https://arxiv.org/abs/2108.13264
>
> Hu et al. 2020 - “Other-Play” for Zero-Shot Coordination - https://proceedings.mlr.press/v119/hu20a/hu20a.pdf

---

> > ### Author Rebuttal · Reviewer_whGx · 2026-04-03
> >
> > Thank you to the authors for fully addressing my concerns. I will keep my score as it is already fairly positive. Great work!

---

> > > ### Author Response · Authors · 2026-04-07
> > >
> > > We sincerely thank the reviewer for the follow-up and are glad that our response addressed your concerns. We really appreciate the positive assessment of the work and all your help and feedback in improving the quality of the work. Thanks again!

---

### Official Review · Reviewer_3isT · 2026-03-10

**Soundness:** 3
**Presentation:** 3
**Significance:** 4
**Originality:** 3
**Overall Recommendation:** 5
**Confidence:** 4

**Summary:**

This paper utilizes the incomplete information game Hanabi as a benchmark to systematically evaluate and explore the multi agent collaborative reasoning capabilities of Large Language Models (LLMs). The authors conduct 2 to 5 player collaboration tests across 17 frontier models, designing three prompting scaffolds of varying complexity, Watson, Sherlock, and Mycroft, to decouple the explicit and implicit state tracking mechanisms of the models. Furthermore, the paper open sources two densely annotated datasets, HanabiLogs and HanabiRewards, upon which a 4B parameter model is fine tuned via Supervised Fine Tuning (SFT) and Reinforcement Learning from LLM Feedback (RLVR). This work validates the efficacy of this fine tuning paradigm in enhancing the collaborative performance and generalization capabilities of lightweight models, while objectively revealing the fundamental limitations of current LLMs in long horizon, multi turn state tracking.

**Compliance With Llm Reviewing Policy:**

Affirmed.

**Final Justification:**

I have read the content carefully and most of my concerns are addressed. I would like to keep the current score and am inclined to accept the paper.

**Key Questions For Authors:**

1.Soft evaluations based on large models lack mathematical verifiability. How do the authors effectively quantify and eliminate the inherent policy biases introduced by the teacher model during the scoring process?

2.Comparing fine tuned models directly against zero shot baselines lacks scientific rigor. Could the authors provide an absolute performance comparison between the 4B model and classical multi agent reinforcement learning algorithms, such as MAPPO, under an equivalent data scale?

3.I recommend precisely quantifying and discussing the token consumption and inference latency under the Mycroft setting.

**Limitations:**

The discussion of limitations in this paper is primarily confined to superficial phenomena such as unstable implicit state tracking, failing to systematically characterize the underlying root causes and extrapolation risks. Since post training relies heavily on strong model trajectories and judge based soft rewards, small models might learn specific policy dialects rather than transferable collaborative principles, subsequently leading to convention drift and coordination mismatch when encountering heterogeneous teammates or out of distribution scenarios. Furthermore, Mycroft style multi round memory and prompting scaffolds incur significant token and latency overheads; however, the paper lacks a quantitative analysis regarding deployment feasibility and cost upper bounds.

**Strengths And Weaknesses:**

Strengths:

Significance: This paper investigates multi agent collaborative reasoning under incomplete information and open sources datasets with dense action annotations, thereby providing highly valuable training and evaluation resources to the community.

Soundness: The experimental design is comprehensive with rigorous variable control; it conducts exhaustive benchmarking across 17 frontier models and effectively decouples and quantifies the explicit and implicit state tracking capabilities of these models utilizing benchmark tests and three tier prompting scaffolds.

Presentation: This paper intuitively illustrates the capability boundaries of various models across different player scales; furthermore, the appendix provides exhaustive prompt templates and error analyses, significantly facilitating research reproducibility.

Weaknesses:

1.Referring to the soft scoring by large models as RLVR lacks rigorousness, as it fundamentally constitutes RLAIF. I recommend revising the terminology to objectively reflect the inherent biases of the judge model.

2.Comparing the absolute scores of small parameter models fine tuned on domain specific trajectories against general large models in a zero shot setting primarily highlights a knowledge distillation effect; moreover, in most scenarios, these fine tuned small models can approach the performance of closed source large models on certain tasks. I suggest supplementing the experiments with classical reinforcement learning baselines of comparable scale to ensure a comprehensive evaluation.

3.The authors should address the prohibitive token consumption and inference latency issues induced by long contexts in complex settings; additionally, I recommend supplementing discussions on the critical issue of convention collapse among heterogeneous agents.

---

> ### Author Rebuttal · Authors · 2026-03-31
>
> We thank the reviewer for their detailed and positive review and are glad they found our work provides **"highly valuable training and evaluation resources"** with a **"comprehensive experimental design and rigorous variable control"** while **"significantly facilitating research reproducibility"**. Below we address the weaknesses (W) and key questions (KQ) raised in the review.
>
> 1. **W1**: We thank the reviewer for this observation. We used the RLVR terminology for consistency with prior work ([*Su et al., 2025*](https://arxiv.org/abs/2503.23829)), as acknowledged in Sec. 3.4 (lines 207–209). We agree that this terminology deserves more precise treatment, and that the recently proposed Rubrics as Rewards (RaR) framework ([*Gunjal et al., 2025*](https://arxiv.org/abs/2507.17746)) which characterizes settings where an LLM evaluates outputs against structured criteria rather than open-ended preferences more accurately describes our approach. We will adopt this terminology in the revised manuscript, with a note clarifying its relationship to the RLVR framing in *Su et al. (2025)*.
>
> 2. **W2 / KQ1**: We agree that part of the gain reflects knowledge transfer from teacher models, but argue that the **results go beyond distillation**:
>
>     i. SFT alone adds +21% improvement, while RL training adds **+156%** (4.8→12.3 in Sherlock, 1.7→8.3 in Mycroft), suggesting the reward signal drives learning beyond imitation.
>
>     ii. Out-of-domain generalization (Tab. 2): improvements on the Group Guessing Game (+11%), EventQA (+6.4%), and IFBench (+1.7 Pass@10), tasks the judge never scored, suggests the model **learned general cooperative reasoning rather than teacher-specific artifacts**. Also, AIME 2025 performance is maintained (Pass@10 = 73.3), indicating no overfitting.
>
>     iii. Teacher Bias: while individual move ratings are soft, the final Hanabi score (0–25) is fully verifiable and judge-independent. Consistent gains across all player settings on held-out seeds (Fig. 7) provide concrete evidence that the ratings carry genuine strategic signal. We also train models with data from different judges (grok3mini and o3) to mitigate backbone-specific biases. We will discuss this further in the revision.
>
> 3. **KQ2**: Following the reviewer’s suggestion, we trained MAPPO baselines using the [official implementation](https://github.com/marlbenchmark/on-policy) for 2–5 player settings. Each MAPPO model was trained on 64K games per player (total 256K) setting which is roughly 3200× more games than the 20 games per player (total 80) setting used for our dataset construction. Below we report average scores over each 6400-game window during training.
>
> | # Games | 2P | 3P | 4P | 5P |
> | ------------- | ---: | ---: | ---: | ---: |
> | 1–6400 | 0.79 | 0.87 | 0.00 | 0.68 |
> | 6401–12800 | 2.83 | 5.98 | **6.74** | **6.19** |
> | 12801–19200 | 7.37 | 4.43 | 3.89 | 4.46 |
> | 19201–25600 | 4.14 | 3.78 | 3.76 | 5.24 |
> | 25601–32000 | 3.05 | 3.65 | 6.62 | 4.44 |
> | 32001–38400 | 5.62 | 3.24 | 5.48 | 4.83 |
> | 38401–44800 | 6.40 | **8.79** | 4.44 | 5.94 |
> | 44801–51200 | 7.17 | 7.43 | 5.60 | 2.87 |
> | 51201–57600 | 3.35 | 4.21 | 4.28 | 5.85 |
> | 57601–64000 | **7.88** | 3.89 | 5.49 | 5.16 |
>
> After 64K games, MAPPO training remains volatile, peaking at ~7–8 Hanabi scores at different # Games. Our 4B model achieves 8.3 (Mycroft) and 12.3 (Sherlock) (Fig. 7) using only ~6,000 samples from ~80 games compared to 50M+ games needed by MAPPO. Moreover, classical MARL agents require retraining per player count and do not generalize out-of-domain; our single checkpoint generalizes across 2–5 players (Fig. 27–28).
>
> 4. **W3 / KQ3**: We thank the reviewer for this suggestion. Below we show inference latency (sec), input tokens and output tokens for oursRL-4B on 8×A100 GPUs, averaged over 10 games per player, all run in parallel. All values are per turn.
>
> | Players | Mycroft (In / Out / Lat) | Sherlock (In / Out / Lat) |
> | ------- | ---: | ---: |
> | 2p | 4945 / 5435 / 150s | 2251 / 4707 / 70s |
> | 3p | 5262 / 6182 / 180s | 2507 / 4710 / 75s |
> | 4p | 5375 / 6713 / 195s | 2601 / 4740 / 80s |
> | 5p | 5605 / 6801 / 200s | 2822 / 5022 / 85s |
>
> With Mycroft, both input (4.9K→5.6K) and output tokens (5.4K→6.8K) grow with player count as working memory expands over time with additional players’ visible hands. With Sherlock, input tokens grow (2.2K→2.8K) but output tokens are stable (~4.7K). Mycroft incurs ~2× higher latency than Sherlock due to larger scratchpad context.  We will include this analysis in the revised Appendix.
>
> In our cross-play setup, we did not observe clear evidence of convention collapse. Instead, performance tended to interpolate smoothly between the self-play results of the two participating models (Fig. 6). We will clarify this point in the revision and include a qualitative cross-play example.
>
> *We sincerely thank the reviewer for the constructive and detailed feedback and are happy to discuss any other concerns towards further improving the paper.*

---

> > ### Author Rebuttal · Reviewer_3isT · 2026-04-02
> >
> > Thank you for your response. I have read the content carefully and most of my concerns are addressed. I would like to keep the current score and am inclined to accept the paper.

---

> > > ### Author Response · Authors · 2026-04-07
> > >
> > > We sincerely thank the reviewer for the follow-up and are glad that our response addressed most of your concerns. Your detailed and thoughtful feedback has been very helpful in improving the presentation and quality of the paper. Thanks again for your support!

---

### Official Review · Reviewer_gyGj · 2026-03-13

**Soundness:** 3
**Presentation:** 3
**Significance:** 3
**Originality:** 3
**Overall Recommendation:** 5
**Confidence:** 3

**Summary:**

This paper studies cooperative reasoning under incomplete information on the Hanabi card game. The authors examine the limit of LLMs on cooperative reasoning by progressively applying different prompting / context-engineering techniques. In addition to self-play cooperative reasoning where members are the same LLM in a team, they also study a cross-play setting, where different LLMs cooperate. Comprehensive benchmark results show that even the best LLM struggles with perfect state tracking in multi-turn play. They also curate two datasets, achieving in-domain improvements and showing potentials to generalize on ood datasets.

**Compliance With Llm Reviewing Policy:**

Affirmed.

**Final Justification:**

The authors have addressed all my concerns, so I'm raising my score from 4 to 5.

**Key Questions For Authors:**

1. For the RL training, which trajectories (rollouts) are used? If I understand it correctly, different players may have different context thus different rollouts. Are they all used for training? Or just a single player's rollouts are used.

**Limitations:**

yes

**Strengths And Weaknesses:**

Strengths:
1. This work provides artifacts that can help facilitate future study of cooperative reasoning.
2. The paper is well written and easy to follow. The experiments are comprehensive.
3. The paper provides insights that may help for future study.

Weaknesses:
1. typo: line 379-381, blank expected in "Sherlockand Mycroftsetups". Same at other places where the macros are used.

---

> ### Author Rebuttal · Authors · 2026-03-30
>
> We thank the reviewer for their review and are glad they found our work **"provides insights"** to **"facilitate future study of cooperative reasoning"** and that the manuscript was **"well-written and easy to follow"**. Below we address the weaknesses (W) and key questions (KQ) raised in the review.
>
> 1. **W1:** We thank the reviewer for catching this formatting issue. We will fix all such formatting errors and typos in the revised version.
>
> 2. **KQ1:** We thank the reviewer for this clarifying question. Each training sample can be understood as follows: the game state (as observed by the current player) serves as the prompt, the list of candidate moves serves as the options, and the move ratings assigned by the judge model serve as the rewards. Since each player has a unique partial observation of the game at their turn, each sample is inherently tied to a single player's perspective. The ~6,000 training samples referenced in Section 5.2 (lines 380–381) **span all player positions across all games, but each individual sample contains only the game state as viewed by the player whose turn it is**. We **train on data from all player positions**, which is important because each player encounters a different distribution of partial-information states throughout the game, exposing the model to greater observation diversity and strengthening generalization across player configurations. This is consistent with our evaluation in Appendix I.2 (Figures 27–28), where the fine-tuned model generalizes across 2–5 player settings using a single checkpoint without position-specific retraining. We will clarify these details in the revised manuscript.
>
> We sincerely thank the reviewer once again for their careful reading and encouraging assessment of the paper’s clarity and relevance to cooperative reasoning with LLMs.
>
> *We would be very happy to discuss any further questions about the work, and would really appreciate an appropriate increase in score if reviewers’ concerns are adequately addressed to facilitate acceptance of the paper.*

---

> > ### Author Rebuttal · Reviewer_gyGj · 2026-04-04
> >
> > Thanks for answering the question. After another walk-through of the paper and based on the feedback of other reviewers, I believe that this is a good work and can benefit the community, especially for those who work on the training of decentralized multi-agent systems. Therefore, I will raise my score from 4 to 5.

---

> > > ### Author Response · Authors · 2026-04-07
> > >
> > > We sincerely thank the reviewer for their reassessment of the paper with the rebuttal and for raising your score. We are very glad that you feel our work will benefit the community. Thanks again for your time and valuable feedback towards improving the quality and presentation of our work!

---

### Decision · Program_Chairs · 2026-04-30

**Decision:**

Accept (regular)

**Comment:**

This paper presents a large-scale evaluation of LLM cooperative reasoning on Hanabi, testing 17 frontier models across 2–5 players with three prompting scaffolds (Watson, Sherlock, Mycroft) that isolate implicit versus explicit state tracking. It releases two valuable resources, HanabiLogs (1,520 trajectories) and HanabiRewards (560 games with dense move-level utilities), and shows that fine-tuning a 4B open model on these datasets improves Hanabi score by up to 156%, reaching within 3 points of a strong proprietary reasoning model and surpassing the best non-reasoning model by 52%. The fine-tuned model also generalizes out of domain to cooperative guessing (+11%), temporal reasoning (+6.4%), instruction following (+1.7 Pass@10), and matches AIME 2025.

All three reviewers recommend Accept after rebuttal.

Consensus strengths: comprehensive benchmark breadth across models and player counts, novel scaffold design that cleanly separates memory mechanisms, useful dataset release with dense rewards, thorough ablations including cross-play and out-of-domain transfer, and strong reproducibility via detailed prompts and error analysis.

Main weaknesses raised: limited evaluation budget (10 games per setting), initial lack of classical RL baseline, imprecise use of RLVR terminology for judge-based rewards, missing quantitative token and latency analysis for long-context scaffolds, and concerns about teacher-model bias and convention collapse in heterogeneous teams.